# Synthesis, Biological Activity and Molecular Docking of Chimeric Peptides Targeting Opioid and NOP Receptors

**DOI:** 10.3390/ijms232012700

**Published:** 2022-10-21

**Authors:** Karol Wtorek, Alessia Ghidini, Luca Gentilucci, Anna Adamska-Bartłomiejczyk, Justyna Piekielna-Ciesielska, Chiara Ruzza, Chiara Sturaro, Girolamo Calò, Stefano Pieretti, Alicja Kluczyk, John McDonald, David G. Lambert, Anna Janecka

**Affiliations:** 1Department of Biomolecular Chemistry, Medical University of Lodz, Mazowiecka 6/8, 92-215 Lodz, Poland; 2Department of Chemistry “G. Ciamician”, University of Bologna, Via Selmi 2, 40126 Bologna, Italy; 3Department of Neuroscience and Rehabilitation, University of Ferrara, 44121 Ferrara, Italy; 4Department of Pharmaceutical and Pharmacological Sciences, University of Padova, Largo Meneghetti 2, 35131 Padova, Italy; 5Istituto Superiore di Sanità, National Center for Drug Research and Evaluation, 00161 Rome, Italy; 6Faculty of Chemistry, University of Wroclaw, 50-383 Wroclaw, Poland; 7Department of Cardiovascular Sciences, University of Leicester, Anaesthesia, Critical Care and Pain Management, Leicester Royal Infirmary, Leicester LE2 7LX, UK

**Keywords:** opioid receptors, nociceptin receptor, chimeric peptides, calcium mobilization assay, antitociceptive test, docking studies

## Abstract

Recently, mixed opioid/NOP agonists came to the spotlight for their favorable functional profiles and promising outcomes in clinical trials as novel analgesics. This study reports on two novel chimeric peptides incorporating the fragment Tyr-c[D-Lys-Phe-Phe]Asp-NH_2_ (**RP-170**), a cyclic peptide with high affinity for µ and κ opioid receptors (or MOP and KOP, respectively), conjugated with the peptide Ac-RYYRIK-NH_2_, a known ligand of the nociceptin/orphanin FQ receptor (NOP), yielding RP-170-RYYRIK-NH_2_ (**KW-495**) and RP-170-Gly_3_-RYYRIK-NH_2_ (**KW-496**). In vitro, the chimeric **KW-496** gained affinity for KOP, hence becoming a dual KOP/MOP agonist, while **KW-495** behaved as a mixed MOP/NOP agonist with low nM affinity. Hence, **KW-495** was selected for further in vivo experiments. Intrathecal administration of this peptide in mice elicited antinociceptive effects in the hot-plate test; this action was sensitive to both the universal opioid receptor antagonist naloxone and the selective NOP antagonist SB-612111. The rotarod test revealed that **KW-495** administration did not alter the mice motor coordination performance. Computational studies have been conducted on the two chimeras to investigate the structural determinants at the basis of the experimental activities, including any role of the Gly_3_ spacer.

## 1. Introduction

A major challenge in opioid peptide research is the development of novel drugs with analgesic properties similar to morphine but devoid of its severe side effects. Opioid analgesia is mediated by three types of so-called classical opioid receptors designated µ, δ and κ (or MOP, DOP and KOP, respectively), with MOP being a predominant target of morphine-like compounds [1,2]. Unfortunately, activation of MOP is also responsible for most of the adverse effects of opioids [3,4,5]. The fourth member of the opioid family is the nociceptin/orphanin FQ (N/OFQ) receptor (NOP) [6]. The endogenous ligand of NOP is the heptadecapeptide N/OFQ, considered a non-classical opioid peptide [7,8]. In contrast to the classical endogenous opioids (endorphins, enkephalins, dynorphins), N/OFQ lacks Tyr^1^, and this difference could be, at least in part, responsible for the different pharmacological profile of this ligand [9]. Analgesic effects mediated by NOP are more complex than those elicited by any other member of the opioid receptor family. Indeed, depending on the route of administration [10] and dosage [11], N/OFQ was shown to produce either anti- or pro-nociceptive effects [10,12,13,14,15]. Recently, the development of potential pain therapeutics based on nociceptin analogs has been increasingly explored [16,17].

The analysis of the three-dimensional crystal structures of all four opioid receptors revealed that the amino acid residues forming the binding pocket of NOP differ from those of MOP, DOP and KOP [9,18,19,20,21]. As a consequence, the hydrophilic and hydrophobic parts of the binding pockets of NOP and classical opioid receptors are different, which may further explain great differences in the binding selectivity of NOP agonists and other opioid ligands, in spite of the high sequence homology between NOP and three classical opioid receptors [22,23].

Decades of research led to the conclusion that activation of only one receptor type is insufficient in terms of new drug candidates, and compounds targeting at least two receptors may produce a more desirable pharmacological profile. Such compounds can be classified either as bivalent or bifunctional ligands [24]. Bivalent ligands contain two distinct pharmacophores, each of them able to interact with a different receptor. They are obtained by joining, directly or by means of a linker, pharmacophores targeting two receptors. In contrast, bifunctional ligands possess one highly integrated pharmacophore that can simultaneously activate two receptors. Such ligands are generally non-selective and often obtained by chance rather than through a planned designing process [25,26]. 

The MOP and NOP receptors are co-localized in the brain structures and share common signaling pathways [27,28]; therefore, the design of MOP/NOP ligands emerged as a promising strategy to circumventing the harmful effects of pure MOP agonists [29,30]. Indeed, some early studies indicated the potential of mixed MOP/NOP ligands to reduce the side effects of pure MOP agonists, in particular the development of tolerance and dependence [31,32]. As reported earlier, heterodimerization of the MOP and NOP receptors [33,34] represents an additional important layer of functional complexity [35]. 

Bifunctional as well as bivalent MOP/NOP ligands are already known. Most likely, the most studied bifunctional non-peptide MOP/NOP ligand is cebranopadol, now used in clinical trials for the treatment of severe chronic and neuropathic pain [16]. Cebranopadol was shown to exert full agonist activity at MOP, almost full agonism at KOP and NOP, and partial agonism at DOP [36,37]. Other mixed MOP/NOP receptor ligands showing promising profiles in preclinical studies include SR16435 [38,39] and AT-121 [40], which in non-human primates produced antinociception with reduced respiratory depression and physical dependence. 

Several peptidic bivalent opioid/NOP ligands have also been obtained, in which the acetylated hexapeptide amide Ac-Arg-Tyr-Tyr-Arg-Ile-Lys-NH_2_ (Ac-RYYRIK-NH_2_) was used as the NOP-activating fragment. This peptide, identified from a combinatorial library [41], has a complex pharmacology. Ac-RYYRIK-NH_2_ is an NOP partial agonist, and its pharmacological activity can vary in different functional assays, depending on their efficiency in the stimulus/response coupling [42,43]. 

In a constant effort to develop analgesic drug candidates with better pharmacological profile and tools allowing us to explore the opioid receptor nature and function, various hybrid ligands have been designed. Here, we report the design of two novel opioid/NOP bivalent chimeras in which the NOP-targeting sequence Ac-RYYRIK-NH_2_ was connected either directly or through a triglycine linker to Tyr-c[D-Lys-Phe-Phe]Asp-NH_2_ (**RP-170**), a cyclopeptide which showed mixed MOP/KOP affinity [44]. Receptor binding and functional profile in the calcium mobilization assay of the hybrid analogs at the opioid and NOP receptors were determined. In the in vivo studies, antinociceptive activity of the selected analog (**KW-495**) was assessed in the mouse hot-plate test. The involvement of opioid and NOP receptors in the antinociceptive action of **KW-495** was studied in the co-administration with the opioid antagonist, naloxone, and NOP receptor antagonist SB-61211. Finally, computational studies were conducted on the two chimeras to investigate the structural determinants at the basis of the experimental activities.

## 2. Results

### 2.1. Design of Opioid/NOP Chimeras

The two hybrid compounds, **KW-495** and **KW-496**, were designed by combining our formerly described cyclic MOP/KOP agonist Tyr-c[D-Lys-Phe-Phe]Asp-NH_2_ (**RP-170**) and the NOP receptor-binding peptide Ac-RYYRIK-NH_2_. The two sequences were connected either directly or through a triglycine spacer (Figure 1). 

The chimeras were synthesized using a solid-phase procedure on MBHA Rink amide resin, using standard Fmoc/*t*Bu chemistry and TBTU/DIEA as the coupling agents. D-Lys^2^ and Asp^5^ were introduced as Fmoc-D-Lys(Mtt)-OH and Fmoc-Asp(O-2PhiPr)-OH, respectively (Figure 1). The synthesis of linear sequences was followed by selective deprotection of the side chains of D-Lys and Asp under mild acidic conditions, and on-resin cyclization with TBTU/DIEA.

After cleavage from the resin with 95% TFA and a cocktail of scavengers, both compounds were purified by semipreparative RP HPLC, and their identity was confirmed by high-resolution mass spectrometry (HR ESI-MS). The purity of the compounds was determined to be ≥95% by analytical RP HPLC. The detailed analytical data of the synthesized peptides are provided in the Appendix A.

### 2.2. Pharmacological Characterization of the Chimeric Peptides In Vitro 

In order to evaluate the affinity of novel chimeras at MOP, DOP, KOP and NOP, radioligand binding assays were performed on membrane preparations from CHO cells overexpressing respective recombinant receptors. The results are summarized in Table 1. The parent **RP-170**, characterized by high MOP and KOP affinity (pKi = 9.21 and 8.53, respectively), was used as a reference opioid ligand. Both chimeras **KW-495** and **KW-496** retained the binding profile of the parent, **RP-170**. A small reduction in their MOP affinity was observed as compared to **RP-170,** and they did not bind to DOP up to the 1 µM concentration. At the KOP, **KW-495** had only slightly lower affinity (pKi = 8.40) than the parent; in contrast, **KW-496** showed higher KOP affinity (pKi = 9.10). At the NOP, the highest, comparable to N/OFQ affinity, was obtained for Ac-RYYRIK-NH_2_ (pKi = 9.29 nM), confirming that this peptide carries a sequence interacting with this receptor. Of the two hybrids, **KW-495** efficiently bound to NOP (pKi = 8.65), while **KW-496** showed reduced NOP binding affinity (pKi = 7.35 nM).

The functional effects of the novel analogs at all three classic opioid receptors and at the NOP receptor were evaluated in the calcium mobilization assay performed in cells co-expressing the respective human recombinant receptors and chimeric G proteins that force Gi receptors to couple with the PLC-IP_3_-Ca^2+^ pathway. This assay has been previously set up and validated for the NOP receptor [45] and was later extended to other opioid receptors [46].

The concentration–response curves were obtained for the parent **RP-170** and Ac-RYYRIK-NH_2_, and for both chimeras (Appendix A). Agonist potencies (pEC_50_) and efficacies (α) of the tested ligands are summarized in Table 2. At the MOP, the parent **RP-170** showed full efficacy and potency (pEC_50_ = 8.93, α = 1.0), even higher than those of the reference MOP agonist EM-2 (pEC_50_ = 8.08, α = 1.00). The chimeric **KW-495** and **KW-496** showed only slightly decreased potency and efficacy. At the DOP, both chimeras were inactive. Consistent with the binding results, both **KW-495** and **KW-496** stimulated calcium release in KOP-expressing cells with high potency and efficacy.

In CHO_NOP+Gαqi5_ cells, the standard agonist N/OFQ evoked a robust concentration-dependent stimulation of calcium release, displaying very high potency and efficacy (pEC_50_ = 9.26, α = 1.00). Ac-RYYRIK-NH_2_ and **KW-495** mimicked the stimulatory effect elicited by N/OFQ, showing lower potencies (pEC_50_ = 7.87 and 7.12, respectively) and efficacies (α = 0.80 and 0.69, respectively). **KW-496** displayed an incomplete concentration–response curve, being able to stimulate calcium mobilization only at 1 µM. The parent **RP-170** was completely inactive (Table 2). 

Summing up, both chimeric peptides displayed slightly reduced but still high binding affinity and agonist activity at MOP and KOP, which could be attributed to the **RP-170** portion. At the NOP receptor, only **KW-495** mimicked the effects of Ac-RYYRIK-NH_2_, although with a circa 6-fold reduced potency. Therefore, **KW-495** was selected for the in vivo experiments.

### 2.3. Antinociceptive Activity 

The antinociceptive activity of **KW-495** was assessed in the hot-plate test. Mice’s response to heat was recorded after intrathecal (i.t.) administration of this analog. The dose–response experiment results are shown in Figure 2. **KW-495** elicited strong, dose-dependent antinociceptive activity with an ED_50_ value (jumping response) of 4.59 (1.17–18.09) nmol. In the dose–response experiments with EM-2, the ED_50_ was 3.15 (1.05–9.51) nmol.

### 2.4. Receptor Antagonist Experiments

To characterize the involvement of opioid and NOP receptors in the antinociceptive action of **KW-495**, co-administration studies with naloxone (a non-selective opioid receptor antagonist) and SB-612111 (an NOP-selective receptor antagonist) were performed. The antinociceptive effect of **KW-495** (6.2 nmol/animal, i.t.) was blocked by both antagonists (1 mg/kg, i.p.), showing the involvement of opioid and NOP receptors in eliciting antinociception (Figure 3). 

### 2.5. Motor Performance Study (Rotarod Test)

The administration of **KW-495** (6.2 nmol/animal, i.t.) did not cause any significant effect on the motor coordination of mice as assessed during the rotarod test when compared to the control group (Figure 4).

### 2.6. Computational Studies 

On resuming the above-described experiments, the parent peptide **RP-170** was a dual MOP/KOP ligand with some preference for MOP (sub-nM vs. nM, respectively). As it turned out, the chimeric peptides **KW-495** and **KW-496** showed reduced affinity for MOP, so that **KW-495** became a dual MOP/KOP ligand with equal nM affinity. Interestingly, **KW-496** gained affinity for KOP, hence becoming a dual ligand with superior affinity for KOP (sub-nM). On the other hand, the chimeric **KW-495** was also active towards NOP, hence representing a mixed MOP/KOP/NOP agonist.

The similar MOP affinity of **KW-495** and **KW-496**, having practically the same pKi values in the nM order of magnitude, likely descends from the Tyr-c[D-Lys-Phe-Phe-Asp]- portion inherited from **RP-170**. The presence of the -Arg-Tyr-Tyr-Arg-Ile-Lys-NH_2_ sequence with or without the 3-Gly linker seems to have little relevance. Previous docking simulations showed that **RP-170** is in fact able to bind effectively to MOP thanks to a series of favorable interactions, including the ionic bond between the protonated amine of Tyr and Asp147^(3.32)^ [47]. Therefore, the interactions of **KW-495** and **KW-496** with MOP have been not investigated further.

On the other hand, other non-obvious experimental pharmacological results aroused some interest, i.e., the increased affinity of **KW-496** towards KOP, and the activity of **KW-495** towards NOP. In this perspective, we performed computational investigations using the deposited X-ray structures of the receptors. In particular, since the chimeric **KW-496** and **KW-495** differ only in the 3-Gly sequence, molecular docking computations seemed appropriate to shed light on any role played by this linker.

Since **KW-495** is a mixed agonist, we first considered the possibility that this hybrid might represent a bivalent ligand that targets the receptor heterodimer. In other words, each portion of the hybrid peptide might bind the cognate receptor simultaneously, rather than targeting two GPCRs that are not linked. However, this possibility was ruled out due to the lack of a suitable spacer between the two bioactive portions of the sequence. Indeed, it has been well-documented that bivalent ligands targeting GPCR dimers must be bridged by a spacer with a length between 16 and 22 heavy atoms [48,49].

*Docking of****KW-495****/KOP**and **KW-496**/KOP.* Hence, the bioactive conformations of **KW-495** and **KW-496** were analyzed with AutoDock 4.0 [50] in a model of the receptor obtained from the deposited X-ray structure of human KOP in a complex with the potent opioid agonist MP1104 (PDB ID 6B73) [51]. It was postulated that the activity of the two compounds at KOP mainly resided in the cyclopeptide portion. For each ligand, independent docking runs were performed, and the results were scored and clustered. Protein–ligand complexes from selected docked poses were minimized in explicit TIP3P water molecules and equilibrated by molecular dynamics. More details can be found in the literature [52] and in the experimental section. The overlay of **KW-495** and **KW-496** into KOP is illustrated in Appendix A, and shows that the two peptide conjugates occupy the same cavity of the receptor, albeit with distinct orientations.

The pose of **KW-495**, Tyr^1^-c[D-Lys-Phe-Phe-Asp]-Arg^6^-Tyr-Tyr-Arg-Ile-Lys^11^-NH_2_, with the best score is shown in Figure 5. As expected, in this structure, the Tyr^1^-c[D-Lys-Phe-Phe-Asp] cyclopentapeptide is deeply buried in the crevice delimited by the TM domains 2, 3 and 5–7, while the C-terminal portion remains external to the TM domain, fully entangled within the bends of the extracellular loop 2 (EL-2), making contact only with the top of the transmembrane helices (TM) 5 and 6.

All details are reported in the Appendix A; for brevity, herein, only the most relevant contacts are discussed. The protonated Tyr^1^ of the cyclopeptide adopts a classic disposition in the message-binding region of the receptor, interacting with Asp138^(3.32)^ carboxylate via an ionic bond; the phenolic side chain makes a hydrogen bond (H-bond) with His291^(6.52)^ [53]. Other stabilizing contacts include Met142, Ile290 and Ile294. As for the C-terminal sequence Arg^6^-Tyr-Tyr^8^-Arg^9^-Ile-Lys^11^NH_2_, the most relevant interactions involve the charged side chains. Arg^6^ makes an ionic bond with Asp223 and H-bonds with Leu212 and Gln213; Tyr^8^ phenol is H-bonded to Arg202; Arg^9^ is H-bonded to Phe214 and Gln213; Lys^11^NH_2_ makes an ionic bond with Asp216 and H-bonds with Ser220 and Leu224 (Appendix A).

In the best pose of **KW-496**, Tyr^1^-c[D-Lys-Phe^3^-Phe^4^-Asp]-Gly_3_-Arg-Tyr-Tyr-Arg-Ile-Lys-NH_2_ (Figure 5), the cyclopentapeptide adopts a different orientation within the TM domains as compared to **KW-495**. The protonated amine of Tyr^1^ maintains the salt bridge with Asp138^(3.32)^ carboxylate, while the phenolic side chain shows interactions with Trp287 (pi-pi), Tyr320 (pi-pi and H-bond Tyr320NH-phenolO) and Ile290 (pi-alkyl). Interestingly, in this pose the message tyramine of the cyclic portion of **KW-496** shows some similarities with that of the cyclic peptide alone, **RP-170**, as calculated in MOP [47]. Albeit not totally unusual for KOP ligands [54], this pose is to some extent an alternative to that of the tetrahydroisoquinoline ring of JDTic and to the docked pose of the ligand dynA(1–8) [55], in which the tyramine phenol groups point towards His291.

The Gly_3_ linker is hosted in a vertical position between the two bends of the EL-2. The Arg^9^-Tyr-Tyr-Arg-Ile-Lys^14^-NH_2_ portion holds on to the top of the helices 2 and 5–7, wrapped within the EL ribbons. All contacts are discussed in detail in the Appendix A. Interestingly, the C-terminal peptide contributes to stabilizing the complex thanks to several meaningful contacts (Appendix A). The guanidinic groups of Arg^9^ and Arg^12^ show ionic bonds with the carboxylates of Asp223 (top of TM-5) and Asp206 (EL-2), respectively; the C-terminal Lys^14^NH_2_ interacts with Ile208(EL-2)C=O (H-bond with NHζ^+^), Val118C=O (H-bond with CONH_2_) and Asn122C=O (H-bond with LysNH). 

**Figure 5 ijms-23-12700-f005:**
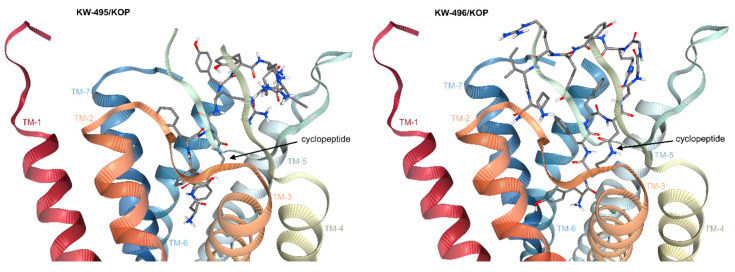
Side views of the predicted complexes of **KW-495**/KOP and **KW-496**/KOP obtained with the receptor models extracted from PDB ID 6B73. The ligands are rendered in thick lines; C is rendered in gray, N in blue and O in red. Figure obtained with PacDOCK web server [56,57].

*Docking of **KW-495**/NOP and KW-496/NOP.* The docking simulations of **KW-495** and **KW-496** into NOP were performed as discussed above. Previous studies indicated that the sequence RYYRIK-NH_2_ competes for the same NOP binding site as N/OFQ [58,59]; therefore, in this study the C-terminal sequence was assumed to be the message portion of the chimeric peptides for NOP. The structure of the receptor was extracted from the crystal structure of NOP in the complex with the ligand C-35, PDB ID 5DHG [60].

In the best-scoring pose of **KW-495**, Tyr1-c[D-Lys-Phe-Phe-Asp]-Arg^6^-Tyr-Tyr-Arg-Ile-Lys^11^-NH_2_, the C-terminal portion Tyr^8^-Arg^9^-Ile-Lys^11^NH_2_ adopts a bent conformation, spanning across the cavity composed by residues of the helices TM-2, 3 and 5–7 (Figure 6). The complex is nicely stabilized by several strong interactions. The guanidinic group of Arg^9^ forms an ionic bond with the carboxylate of Asp138^(3.32)^, and a H-bond with Cys200C=O. The branched side chain of Ile^10^ nicely fits in a hydrophobic cavity shaped by elements of TM-5 and 6 (Val283, Val279 and Leu301). The C-terminal Lys^11^NH_2_ is involved in interactions with Gln107C=O (H-bond), and the carboxylate side chains of both Glu199 and Asp110 (NHζ^+^, ionic bond). Full details are given in the Appendix A.

The cyclopeptide scaffold Tyr^1^-c[D-Lys-Phe-Phe-Asp^5^] completely lies against the top of TM-2 and extensive portions of EL-1 and EL-2 (Appendix A). The N-terminal TyrNH_3_^+^ forms strong interactions with residue of the extracellular loops, i.e., Glu197carboxylate (EL-2, salt bridge), Gly114C=O (EL-1, H-bond) and Leu113C=O (EL-1, H-bond), while the phenol ring of Tyr also interacts with Glu197carboxylate (pi-anion). 

In the best pose of **KW-496**, Tyr^1^-c[D-Lys-Phe^3^-Phe^4^-Asp]-Gly_3_-Arg^9^-Tyr-Tyr-Arg^12^-Ile-Lys^14^-NH_2_ (Figure 6), the C-terminal portion Tyr^11^-Arg^12^-Ile-Lys^14^NH_2_ goes much deeper into the receptor cavity so that the positively charged side chains of Arg^12^ and Lys^10^ form an electrostatic clamp on Asp130^3:32^. The Arg^12^ guanidinic group is also H-bonded with Gln107; Lys^14^NH_2_ is H-bonded with Thr305OH and with Arg^12^C=O (intramolecular); Tyr^11^ interacts with Leu301 (pi-alkyl), Val298 (pi-alkyl and C-H) and Arg^9^ (intramolecular, cation-pi); Tyr^10^C=O interacts with Arg302 guanidine (H-bond and cation-pi); Arg^9^C=O is H-bonded to Glu199(EL-2). Notably, the Gly_3_ linker appears fully exposed outside the receptor. The cyclopeptide ring is weakly hosted between the ELs 2 and 3, and the top of TM-5, but the N-terminal Tyr^1^ fills a deep pocket delimited by TM helices 4 and 5 (Appendix A).

## 3. Discussion

Opioid and NOP receptors are highly expressed in the CNS and play important roles in the direct and indirect control of pain transmission [9,17,61]. The agonists of the classical opioid receptors all produce strong analgesic effects. N/OFQ, when injected intracerebroventricularly (i.c.v.), was shown to display nociceptive activity, suggesting an anti-opioid profile, leading to the assumption that NOP antagonists should have antinociceptive activity [62]. However, other studies reported that N/OFQ potentiated morphine-induced analgesia when injected into the spinal cord, drawing into question the anti-opioid activity of this ligand [63].

With its ambiguous pharmacological profile, N/OFQ appeared as an interesting ligand for use in the design of hybrid molecules. However, due to the long sequence of N/OFQ, a much shorter synthetic peptide, i.e., the NOP ligand Ac-RYYRIK-NH_2_, was preferred as a fragment that would be convenient to combine with MOP or DOP agonists, yielding potential bivalently active opioid/NOP compounds. Kawano et al. [64] joined this peptide with the MOP agonist, dermorphin (Tyr-D-Ala-Gly-Tyr-Pro-Ser), or its shorter analog Tyr-D-Ala-Phe-β-Ala. MOP and NOP ligands were connected tail-to-tail by various spacers. The so-obtained dermorphin-RYYRIK-NH_2_ hybrids retained high binding affinities for both receptors, which increased with the length of the spacer. The authors concluded that longer spacers prevented the interference of the closely located pharmacophores with their respective binding sites. In vivo, the hybrids showed potent and dose-dependent antinociceptive activities in the tail-flick test in mice following i.t. administration [58].

Guillemyn et al. [31] linked Ac-RYYRIK-NH_2_ to the opioid ligand Dmt-D-Arg-Aba-β-Ala-NH_2_ (Dmt: 2′,6′-dimethyl-L-tyrosine; Aba = tetrahydro-4-amino-2-benzazepinone). The resulting hybrid, Dmt-D-Arg-Aba-β-Ala-Arg-Tyr-Tyr-Arg-Ile-Lys-NH_2_ (KGNOP1), showed decreased affinity to opioid and NOP receptors, behaving as an agonist at MOP and an antagonist at NOP. In rats, KGNOP1 produced a strong analgesic effect after i.t. administration, but repeated daily injections led to the development of tolerance [65]. Further in vivo studies showed that in the formalin test, KGNOP1 was highly potent and efficacious in rodent models of acute nociceptive, acute and chronic inflammatory pain and chronic neuropathic pain after systemic (i.v. or s.c.) administration. Antinociceptive effects of KGNOP1 were reversed by naltrexone and SB-612111, indicating the involvement of both MOP and NOP receptor agonism. This is in contrast to the in vitro data indicating antagonism at NOP [66].

Recently, Erdei et al. [59,67] described three hybrid ligands, combining Ac-RYYRIK-NH_2_ with N-terminal tetra- or pentapeptide sequences of DOP and KOP agonists, i.e., Leu-enkephalin and dynorphin. Both fragments were linked either directly or through a 3-Gly spacer. The YGGF-RYYRIK-NH_2_ analog deprived of a spacer lost most of the DOP-mediated activity but exhibited the most efficacious agonist action in the MOP-mediated G-protein stimulation assay. 

In this context, we decided to design chimeric peptides composed of **RP-170,** a mixed MOP/KOP cyclic peptide ligand with a preference for MOP, and the NOP active Ac-RYYRIK-NH_2_ (Figure 1)_._ Herein, we report the synthesis and the initial pharmacological testing of two chimeras in which **RP-170** and -RYYRIK-NH_2_ were linked directly (**KW-495**) or through a 3-Gly spacer (**KW-496**).

In vitro binding and functional assays showed that **KW-496** became a mixed ligand with inverted KOP > MOP affinity, without activity at NOP. On the other hand, **KW-495** could simultaneously and potently bind and activate opioid and NOP receptors. In vivo, **KW-495** produced a dose-dependent antinociceptive effect when given i.t., and this effect was antagonized by naloxone and SB-612111. This finding demonstrates the involvement of both classical opioid and NOP receptors in the antinociceptive action of **KW-495.** Finally, this peptide did not alter the locomotor activity, and there was no change in the motor coordination and balance of mice.

Taken together, these findings demonstrate that the cyclic opioid **RP-170** and the NOP ligand Ac-RYYRIC-NH_2_ can be successfully joined to generate a chimeric peptide able to simultaneously target NOP and classical opioid receptors.

Computational simulations of the chimeric peptides docked in the receptors may help us in understanding the structural determinants at the basis of the comparatively higher KOP activity of **KW-496**, and the significant NOP activity of **KW-495**. In **KW-495**, the NOP-active sequence RYYRIK-NH_2_ is directly connected to the MOP/KOP ligand **RP-170**, while **KW-496** possesses a Gly_3_ sequence in between.

In **KW-495**/KOP, the protonated Tyr^1^ of the cyclopeptide is predicted to adopt the classic disposition within the message-binding region of the receptor, forming the expected ionic bond with Asp138^(3.32)^ carboxylate and a hydrogen bond (H-bond) with His291^(6.52)^; the C-terminal RYYRIK-NH_2_ portion remains external to the TM domain, fully entangled within the bends of EL-2.

In contrast, the presence of a Gly_3_ spacer in **KW-496**/KOP is predicted to allow a different position of the cyclopeptide and of the *C*-terminal RYYRIK-NH_2_. The computations suggest that the spacer spans vertically across the transmembrane domain. This thin and flexible connection seems to allow the *N*-terminal cyclopentapeptide to adopt an alternative orientation (Figure 5 and Appendix A). In this structure, the tyramine message portion of the cyclic peptide of **KW-496** maintains in KOP a similar disposition to that adopted by the tyramine of the cyclic peptide **RP-170** alone as calculated in MOP [47]. This significant difference might in part explain the higher experimental affinity of **KW-496** as compared to **KW-495.** In addition, the residues of the C-terminal section can attain further, highly favorable interactions with the residues at the outer side of TM2 and the EC loops (Figure 5).

The significant NOP activity of **KW-495**, deprived of any linkers, was somewhat unexpected. Kawano et al. [64] joined the MOP agonist dermorphin to RYYRIK-NH_2_ with various spacers, and observed that binding to both receptors increased with the length of the spacer. The authors proposed that longer spacers allowed avoiding any interference of the closely located pharmacophores with their respective binding sites.

The structure of **KW-495**/NOP is characterized by a transversal, bent conformation of the *C*-terminal tetrapeptide (Figure 6 and Appendix A). In 2012, Daga and Zaveri used the homology model of NOP to analyze the binding of Phe-Gly-Gly-Phe-Thr-Gly-Ala-Arg-Lys-Ser-Ala-Arg-Lys, N/OFQ (1–13). These authors proposed that the *N*-terminal tetrapeptide Phe-Gly-Gly-Phe of N/OFQ penetrates deeply into the binding pocket in a straight conformation, and the N-terminal amine makes an essential charge interaction with Asp130 [68]. Very recently, Dumitrascuta et al. simulated NOP binding of the chimeric peptide Dmt^1^-D-Arg-Aba-β-Ala-Arg-Tyr-Tyr-Arg^8^-Ile-Lys^10^NH_2_ (KGNOP1), a full agonist for both MOP and NOP [66]. Similar to **KW-496**, the computations performed by Dumitrascuta et al. showed an extended conformation of the *C*-terminal tetrapeptide, with Arg^8^ and Lys^10^ forming salt bridges with Asp130^3:32^, while the linear *N*-terminal tetrapeptide was excluded from the receptor, being fully exposed to the solvent. In contrast to KGNOP1, the N-terminal portion of **KW-495** external to the receptor is represented by a large cyclopeptide scaffold. The rigidity and the noteworthy impediment exerted by the latter seem to prevent the *C*-terminal sequence from fully plunging into the receptor’s depths. Nevertheless, the computations predict that the flexible *C*-terminal sequence is still capable of establishing strong interactions with NOP, while the cyclopeptide is tightly pressed against the outside of the receptor, and significantly contributes to overall binding with several strong interactions (Appendix A).

## 4. Materials and Methods

### 4.1. General Methods

Most of the chemicals and solvents were obtained from Sigma Aldrich. Protected amino acids were provided by Trimen Co. (Lodz, Poland) and the MBHA Rink-Amide peptide resin (100–200 mesh, 0.8 mmol/g) by NovaBiochem. Opioid radioligands [^3^H]DAMGO, [^3^H]deltorphin-2 and [^3^H]U-69593 and human recombinant opioid receptors came from PerkinElmer (Krakow, Poland). GF/B glass fiber strips were purchased from Whatman (Brentford, UK). Analytical and semi-preparative RP HPLC was performed using a Waters Breeze instrument (Milford, MA, USA) with a dual absorbance detector (Waters 2487). The HR ESI-MS experiments were performed on a Shimadzu LCMS-IT-TOF (ion trap–time-of-flight) hybrid mass spectrometer (Shimadzu, Japan) equipped with an ESI source connected to a Nexera HPLC system (Shimadzu, Japan) with auto tuning in the positive-ion mode. Peptide solutions (1 μL) were introduced in a 0.2 mL/min flow of mobile phase (water:acetonitrile (1:1) with 0.1% HCOOH).

### 4.2. Peptide Synthesis

Peptides were synthesized by the standard solid-phase procedure on MBHA Rink-Amide peptide resin using the Nα-Fmoc strategy. The Nα-amino group of D-Lys was protected by 4-methyltrityl (Mtt), the β-carboxy group of Asp by 2-phenyl-isopropyl ester (O-2PhiPr) and the hydroxy group of Tyr by t-butyl (t-Bu). Piperidine in DMF (20%) was used for the deprotection of Fmoc groups, 2-(1H-benzotriazol-1-yl)-1,1,3,3-tetramethyluronium tetrafluoroborate (TBTU) was employed as a coupling agent, and diisopropylethylamine (DIEA) was used as a neutralizing base. Fully assembled Fmoc-protected linear peptides were treated with 1% trifluoroacetic acid (TFA) in dichloromethane (DCM) to remove the side chain Mtt and O-2PhiPr-protecting groups, followed by on-resin cyclization (TBTU). Cleavage from the resin was accomplished by the treatment with TFA/triisopropylsilane (TIS)/water (95:2.5:2.5) for 3 h at room temperature. 

Crude peptides were purified by preparative reversed-phase HPLC on a Vydac C18 column (10 μm, 22 mm × 250 mm), flow rate 2 mL/min, 20 min linear gradient from water/0.1% (*v/v*) TFA to 80% acetonitrile/20% water/0.1% (*v/v*) TFA. The purity of the final peptides was verified by analytical HPLC employing a Vydac C18 column (5 μm, 4.6 mm × 250 mm), flow rate 1 mL/min and the same solvent system over 50 min. The purity of the obtained peptides was >95%. Calculated values for protonated molecular ions were in agreement with those determined by high-resolution mass spectroscopy with electrospray ionization (ESI-MS) (Appendix A). 

### 4.3. Cell Culture

Transfected cell lines were maintained in culture medium consisting of Dulbecco’s MEM/HAM’S F-12 (50/50) supplemented with 10% fetal bovine serum (FBS) and streptomycin (100 μg/mL), penicillin (100 IU/mL), l-glutamine (2 mmol/L), geneticin (G418; 200 μg/mL), fungizone (1 μg/mL) and hygromycin B (100 μg/mL). Cell cultures were kept at 37 °C in 5% CO_2_ humidified air. When confluence was reached (3–4 days), cells were sub-cultured as required using trypsin/EDTA and used for testing.

### 4.4. Membrane Preparation

CHO cells (a kind gift from Dr F. Marshall and Mrs N. Bevan, GSK, Stevenage, UK) stably expressing the human nociceptin/orphanin FQ peptide receptor (CHO_hNOP_) were cultured in Dulbecco’s Modified Eagles Medium/Nutrient Mixture F12 Ham (50/50) supplemented with 5% fetal calf serum, penicillin (50 IU/mL), streptomycin (100 µg/mL), fungizone (2.5 µg/mL) and L-glutamine. Stock cell culture media were further supplemented with geneticin (G418) (2 µg/mL), hygromycin-B (2 µg/mL) and cells grown at 37 °C in 5% carbon dioxide humidified air. Cells were used at confluence and membrane fragments prepared by homogenization of cells followed by centrifugation at 13,500 rpm for 10 min at 4 °C. This was repeated for a total of three times [69]. Protein concentration was determined using the method described by Lowry [70].

### 4.5. [^3^H]DAMGO, [^3^H]deltorphin-2, [^3^H]U-69593 Displacement Binding Assay

Commercial membranes of CHO cells (Perkin Elmer, Inc., Waltham, MA, USA) stably expressing human opioid receptors and the competing radioligands, [^3^H]DAMGO, [^3^H]deltorphin-2 and [^3^H]U-69593 for MOP, DOP and KOP, respectively, were used. Membranes were incubated in 0.5 mL of 50 mM Tris/HCl (pH = 7.4), 0.5% bovine serum albumin (BSA), with a number of peptidase inhibitors (bacitracin, bestatin, captopril) and various concentrations of radioligands for 2 h at 25 °C. Non-specific binding was assessed in the presence of 10 mM naloxone. Assays were incubated at room temperature for 1 h and reactions were terminated via filtration through Whatman GF/B filters, soaked in 0.5% polyethyleneimine (PEI), using Millipore Sampling Manifold (Billerica, MA, USA). Radioactivity was determined after an 8 h extraction period [69].

### 4.6. [Leucyl-^3^H]nociceptin Displacement Binding Assay

CHO_hNOP_ cell homogenate (10–40 µg) was incubated in 0.5 mL of buffer containing Tris-HCl (50 mM), MgSO_4_ (5 mM, pH = 7.4) containing bovine serum albumin (0.5%), [leucyl-^3^H]nociceptin (approximately 0.3 nM) and varying concentrations of the tested ligands (**RP-170**, **KW-495**, **KW-496** and Ac-RYYRIK-NH_2_) and N/OFQ control. Non-specific binding was determined in the presence of 1 µM N/OFQ. Assays were incubated at room temperature for 1 h and reactions were terminated via filtration through Whatman GF/B filters, soaked in 0.5% polyethyleneimine (PEI), using a Brandel harvester. Radioactivity was determined after an 8 h extraction period [69].

### 4.7. Binding Data Analysis

The data were analyzed by a nonlinear least square regression analysis computer program Graph Pad PRISM 6.0 (Graph Pad Software Inc., San Diego, CA, USA). The IC_50_ values were determined from the logarithmic concentration–displacement curves, and the values of the inhibitory constants (K_i_) were calculated according to the equation of Cheng and Prusoff [71].

### 4.8. Calcium Mobilization Functional Assay

For the experiments, CHO cells (a kind gift from Prof. T. Costa, Rome, Italy) stably co-expressing the human MOP, DOP, KOP or NOP and the C-terminally modified G-protein were generated as described [46]. Briefly, cells incubated for 24 h in 96-well black, clear-bottom plates were loaded with medium supplemented with probenecid (2.5 mmol/L), calcium-sensitive fluorescent dye Fluo-4 AM (3 μmol/L) and pluronic acid (0.01%) and kept for 30 min at 37 °C. Following aspiration of the loading solution and a washing step, serial dilutions of peptide stock solutions were added. After placing cell culture and compound plates into the FlexStation II (Molecular Devices, Sunnyvale, CA, USA), changes in fluorescence of the cell-loaded calcium-sensitive dye Fluor-4 AM were measured. On-line additions were carried out in a volume of 50 μL/well.

### 4.9. Functional Assay Data Analysis and Terminology

All data were analyzed using Graph Pad Prism 6.0 (La Jolla, CA, USA). Concentration–response curves were fitted using the four-parameter log-logistic equation. Data are expressed as mean ± SEM of at least four experiments performed in duplicate. Agonist effects were expressed as maximum change in percent over the baseline fluorescence. Baseline fluorescence was measured in wells treated with vehicle. Agonist potency was expressed as pEC_50_, which is the negative logarithm to base 10 of the agonist molar concentration that produces 50% of the maximal possible effect of that agonist. Concentration–response curves to agonists were fitted with the four-parameter logistic nonlinear regression model:(1)$$\mathrm{Effect}=\mathrm{Baseline}+\frac{\left(E_{\max}-\mathrm{Baseline}\right)}{\left(1+10^{({\mathrm{LogEC}}_{50}-{\;\mathrm{Log}}_{\left(\mathrm{compound}\right)}\mathrm{Hillslope}}\right)}$$

### 4.10. In Vivo Studies

#### 4.10.1. Animals

Male CD-1 mice (Charles River, Italy), weighing 25–30 g, were used for the study. The animals were housed in a room with controlled temperature (21 ± 1 °C), humidity (70 ± 5%) and light/dark cycle conditions (12/12 h), with free access to laboratory chow and tap water. Mice were housed for at least 1 week before the experimental sessions in sawdust-lined plastic cages (6 mice in each cage). Animal studies and research protocols were approved by the Service for Biotechnology and Animal Welfare of the Istituto Superiore di Sanità and authorized by the Italian Ministry of Health, according to Legislative Decree 26/14 (protocol number: 198/2013-B), which implemented the European Directive 2010/63/EEC on laboratory animal protection. Animal welfare was routinely checked by veterinarians from the Service for Biotechnology and Animal Welfare.

#### 4.10.2. Drugs and Pharmacological Treatment

The tested compounds were dissolved in DMSO and further diluted with 0.9% NaCl solution to the final concentration of 0.1% DMSO. Animals without treatment (control group) received the vehicle alone (0.1% DMSO in 0.9% NaCl solution). The vehicle given alone had no effects on the observed parameters.

#### 4.10.3. Assessment of Antinociception

The antinociceptive effects of peptides were assessed in the hot-plate test in mice as described earlier [72]. The intrathecal (i.t.) injections (5 μL/animal) of peptide or vehicle were performed under inhalational isoflurane anesthesia using the method described by Hylden and Wilcox [73]. Injections were performed with disposable, 30-gauge, ½-inch needles mated to a 10 μL syringe. Mice were held firmly by the pelvic girdle, while the needle was introduced at an angle of about 20° above the vertebral column. The needle was inserted between the L5 and L6 spinous processes and moved carefully forward to the intervertebral space as the angle of the syringe was decreased to about 10°. The tip of the needle was inserted approximately 0.5 cm into the vertebral column. Antagonists (naloxone hydrochloride or SB-612111 hydrochloride) were intraperitoneally (i.p.) administered 15 min prior to the peptide or vehicle injection at the dose of 1 mg/kg. A transparent plastic cylinder (14 cm diameter, 20 cm height) was used to confine a mouse on the heated (55 ± 0.5 °C) surface of the plate (Ugo Basile, No. 7280, Gemonio, Italy). The animals were placed on the hot plate 5 min after the i.t. injection of saline (control) or peptide, and the latencies to jumping were measured at 5, 10, 20, 30, 45 and 60 min after administration of peptide. A cut-off time of 120 s was used to prevent tissue injury. The percentage of the maximal possible effect (%MPE) was calculated as: %MPE = (t1 − t0)/(t2 − t0) × 100, where t0: control latency, t1: test latency, and t2: cut-off time. The median antinociceptive dose (ED50) was calculated according to the method of Litchfield and Wilcox [74]. The data were analyzed by a nonlinear least-square regression analysis computer program, Graph Pad PRISM 6.0 (GraphPad Software Inc., San Diego, CA, USA).

#### 4.10.4. Assessment of Motor Coordination

Rotarod apparatus (Ugo Basile, No. 8466, Italy) was used to estimate the probable influence of peptide on motor coordination of animals [75]. In the training period, 3 trials were conducted for 3 consecutive days. Animals that were able to stay on the rotating (16 revolutions/min) bar more than 180 s were chosen for the tests. In the experimental session, each animal was tested on the rotarod, and duration to fall from the rotating bar was noted.

### 4.11. Molecular Docking

The KOP model was obtained from the deposited X-ray structure PDB ID 6B73, while the NOP model was obtained from the crystal structure PDB ID 5DHG. The initial structures of the ligands were optimized using the program ORCA [76] with a Hartree-Fock method and a 6-31G* basis set.

The simulations were performed with AutoDock 4.0, using the Lamarckian genetic algorithm, which combines global (genetic algorithm) and local (Solis and Wets algorithm) search. Ligands and receptors were further processed using the AutoDock Tool Kit (ADT). Gasteiger–Marsili charges were loaded on the ligands in ADT and solvation parameters were added to the final structure using the Addsol utility. The docking runs were applied on an initial population of 100 random structures, with a maximum number of energy evaluations = 150, mutation rate = 0.02, a crossover rate = 0.80, elitism = 1.25. The so-called pseudo-Solis and Wets algorithm was applied to local search, with the maximum of iterations per local search per ligand = 200. The grid maps representing the system in the actual docking process were calculated with Autogrid. Grid dimensions were 80 × 80 × 80, and the spacing between the grid points and the center of the cavity left by the ligand after its removal was set to 0.1 Å. Structures differing < 1.0 Å in positional root mean square deviation (RMSD) were clustered together. For each cluster, the structure with the most favorable free energy of binding was selected for subsequent minimization of the protein–ligand complexes.

The selected ligand–protein complex structures were prepared by assigning AM1-BCC charges [77] to the ligands and AMBER 14SB force field [78] parameters to the proteins. Moreover, the complexes were solvated with TIP3P water molecules. Additional Na^+^ and Cl^-^ ions were placed in the water box to obtain a physiological ion strength (200 mM). Systems were minimized and equilibrated at 300 K and 1 atm by performing 1 ns MD simulations with position restraints on the protein backbone.

## Figures and Tables

**Figure 1 ijms-23-12700-f001:**
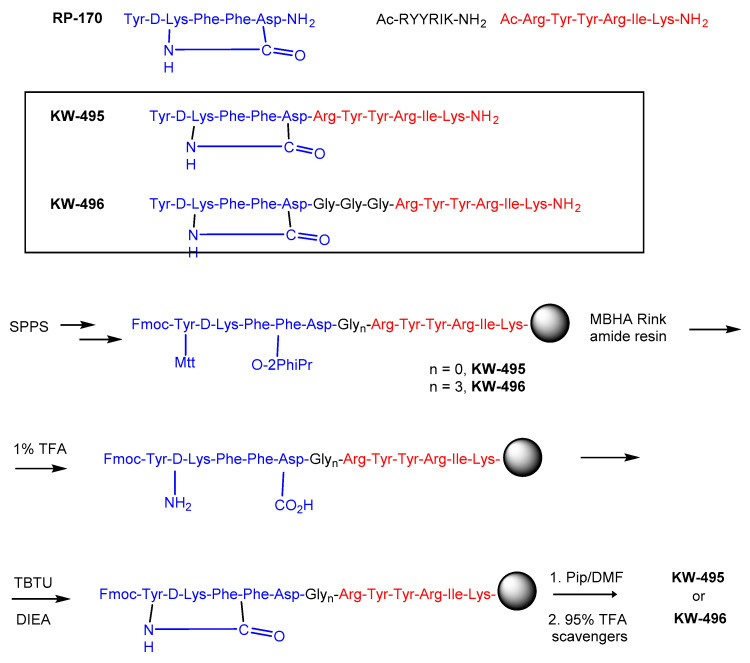
Structures of the parent peptides **RP-170** and Ac-RYYRIK-NH_2_ and of the hybrid derivatives **KW-495** and **KW-496**, and the synthetic strategy.

**Figure 2 ijms-23-12700-f002:**
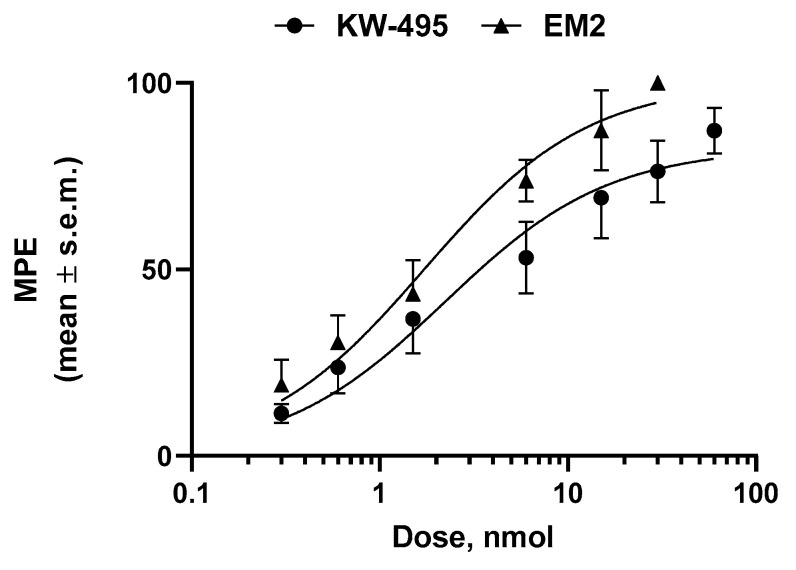
Effect of different doses of **KW-495** chimera and EM-2 in the mouse hot-plate test. Results are expressed as percentage (mean ± SEM) of the maximal possible effect (%MPE) for the inhibition of jumping induced by i.t. injection of **KW-495** or EM-2. *n* = 6–10 mice for each experimental group.

**Figure 3 ijms-23-12700-f003:**
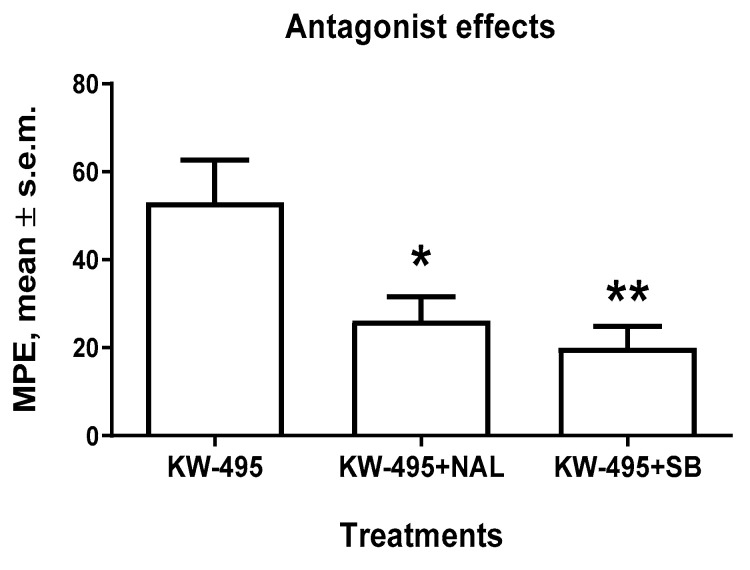
Antagonist effect of naloxone hydrochloride (NAL) or SB-612111 hydrochloride (SB), both at 1 mg/kg i.p., on the inhibition of jumping by administration of **KW-495** (6.2 nmol/animal, i.t.) in the mouse hot-plate test. Results are expressed as percentage (mean ± SEM) of the maximal possible effect (%MPE). *n* = 10–12 mice for each experimental group. Statistical significance was assessed using one-way ANOVA followed by Dunnett’s multiple comparisons test. * is for *p* < 0.05 and ** is for *p* < 0.01 as compared with **KW-495**.

**Figure 4 ijms-23-12700-f004:**
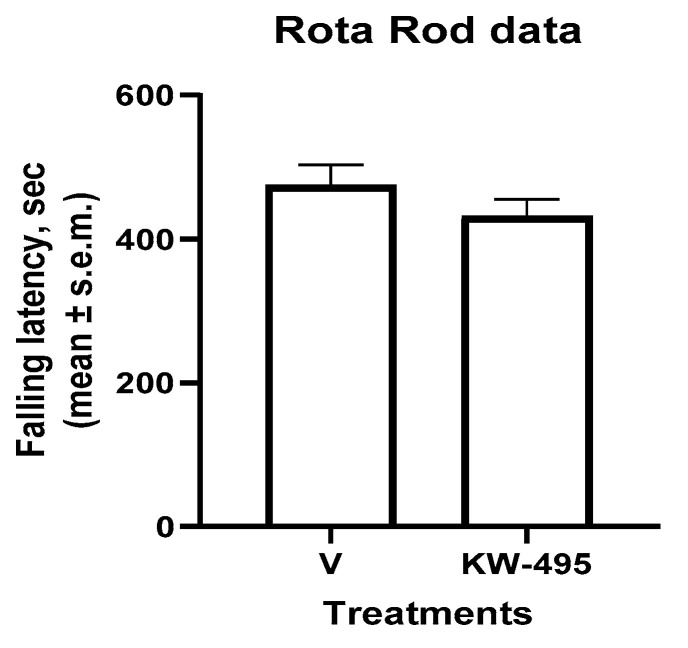
Effects of **KW-495** (6.2 nmol/animal, i.t.) on falling latencies of mice in the rotarod test, *n* = 7–8 mice for each experimental group.

**Figure 6 ijms-23-12700-f006:**
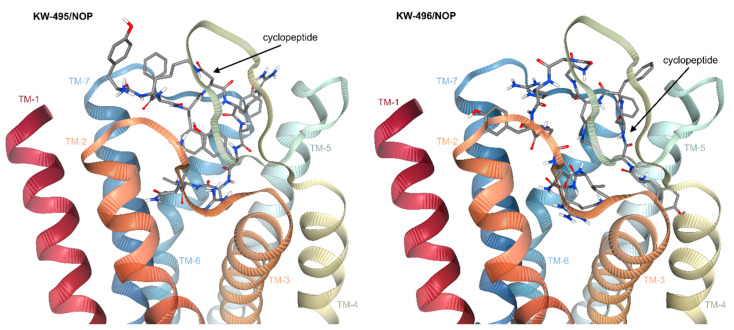
Side views of the predicted complexes of **KW-495**/NOP and **KW-496**/NOP obtained with the receptor models extracted from PDB ID 5DHG. The ligands are rendered in thick lines; C is rendered in gray, N in blue and O in red. Figure obtained with PacDOCK web server [56,57].

**Table 1 ijms-23-12700-t001:** Binding affinities of reference compounds and hybrid peptides at classical opioid and nociceptin receptors.

Compd.	Sequence	pK_i_			
		MOP	DOP	KOP	NOP
**EM-2**	Tyr-Pro-Phe-Phe-NH_2_	9.14 ± 0.08	<6	<6	N.D.
**N/OFQ**	Phe-Gly-Gly-Phe-Thr-Gly-Ala-Arg-Lys-Ser-Ala-Arg-Lys-Leu-Ala-Asn-Gln	N.D.	N.D.	N.D.	9.09 ± 0.24
**Ac-RYYRIK-NH_2_**	Ac-Arg-Tyr-Tyr-Arg-Ile-Lys-NH_2_	N.D.	N.D.	N.D.	9.29 ± 0.85
**RP-170**	Tyr-c[D-Lys-Phe-Phe-Asp]-NH_2_	9.21 ± 0.05	6.53 ± 1.21	8.53 ± 0.09	N.D.
**KW-495**	Tyr-c[D-Lys-Phe-Phe-Asp]-Arg-Tyr-Tyr-Arg-Ile-Lys-NH_2_	8.35 ± 1.11	<6	8.40 ± 2.12	8.65 ± 0.42
**KW-496**	Tyr-c[D-Lys-Phe-Phe-Asp]-Gly-Gly-Gly-Arg-Tyr-Tyr-Arg-Ile-Lys-NH_2_	8.33 ± 0.79	<6	9.10 ± 0.87	7.35 ± 0.13

Displacement of [^3^H]DAMGO, [^3^H]deltorphin-2, [^3^H]U-69593 and [*leucyl*-^3^H]nociceptin from CHO_hMOP_, CHO_hDOP_, CHO_hKOP_ and CHO_hNOP_ cell homogenates, respectively. Data are mean ± SEM for *n* = 5. N.D.—not determined.

**Table 2 ijms-23-12700-t002:** Effects of the hybrids and of reference compounds at human recombinant opioid receptors coupled with calcium signaling via chimeric G proteins. EM-2, DPDPE, dynorphin A and N/OFQ were used as reference agonists for calculating intrinsic activity at MOP, DOP, KOP and NOP, respectively.

Peptide	MOP		DOP		KOP		NOP	
	pEC_50_ ^a^ (CL95%)	α ^b^ ± SEM	pEC_50_ (CL95%)	α ± SEM	pEC_50_ (CL95%)	α ± SEM	pEC_50_ (CL95%)	α ± SEM
**EM-2**	8.08 ± 0.06	1.00	inactive ^c^		inactive		inactive	
**DPDPE**	inactive		7.23 ± 0.22	1.00	inactive		inactive	
**dynorphin A**	6.67 ± 0.50 ^a^	0.83 ± 0.10	7.73 ± 0.27	1.00	8.78 ± 0.05	1.00	inactive	
**N/OFQ**	inactive		inactive		inactive		9.26 ± 0.45	1.0
**RP-170**	8.93 ± 0.05	1.00	inactive		8.60 ± 0.14	1.00 ± 0.03	inactive	
**Ac-RYYRIK-NH_2_**	inactive		inactive		inactive		7.87 ± 0.49	0.80 ± 0.06
**KW-495**	8.06 ± 0.46	0.83 ± 0.07	inactive		8.55 ± 0.17	1.02 ± 0.03	7.12 ± 0.13	0.69 ± 0.10
**KW-496**	8.31 ± 0.30	0.95 ± 0.07	inactive		8.94 ± 0.18	1.03 ± 0.02	Crc incomplete at 1 μM, 0.33 ± 0.10	

^a^ Agonist potency values (pEC_50_). ^b^ Efficacy values (α). ^c^ Inactive means that the compound was inactive up to 1 μM. *p* < 0.05 according to one-way ANOVA followed by the Dunnett test for multiple comparisons.

## Data Availability

PacDOCK is freely available at https://pegasus.lbic.unibo.it/pacdock (accessed on 18 October 2022).

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
