# Peer review of "Synthesis, Biological Activity and Molecular Docking of Chimeric Peptides Targeting Opioid and NOP Receptors"

_ijms, 2022, doi:10.3390/ijms232012700_

Round 1

Reviewer 1 Report

Wtorek et al developed two chimeric peptides targeting opioid receptors by direct or linker fusion of MOP/KOP and NOP agonists. In vitro binding assay, antinociceptive activity, receptor antagonist experiments, and motor performance studies were carried out to prove the function of the chimeric peptides. Both peptides showed high affinity toward MOP/KOP and NOP, with significant antinociceptive activity. Docking experiments were also performed targeting KOP and NOP.

The experimental data is solid and convincing. However, the docking results can only be used for reference. It would be more reasonable to start with docking of building blocks (RYYRIK and cyclic peptide) to determine the exposed part of each block when bound to receptor, then design conjugates based on the docking conformations.

Some minor points:

Line 124 pKi = 9.29 (no nM)

What would happen if mutations were made on the chimeric peptide? (eg mutating residues on RYYRIK motif or Tyr or cyclic KFFD)

It would be interesting for the authors to show an overlay of KW-495 and KW-496 binding to KOP. From Figure 5 it seems that the identical cyclic peptide binds to KOP in distinct conformations. This might indicate that the docking results are not reliable. Are the structures in agreement with ref 48?

Docking of RYYRIK peptide alone should also be performed to validate the docking results of KW-495 and KW-496.

What about the docking scores? Are they consistent with the binding data?

Author Response

The experimental data is solid and convincing. However,...

Comment. … the docking results can only be used for reference. It would be more reasonable to start with docking of building blocks (RYYRIK and cyclic peptide) to determine the exposed part of each block when bound to receptor, then design conjugates based on the docking conformations.

Answer. We are aware that the calculated structures represent mere cues for the discussion of the pharmacological data. We the thank the reviewer for the interesting suggestion for future design, i.e. to start from the docking of distinct fragments. In this work we used a more traditional strategy; initially, we designed the hybrids, and performed biological studies, then we performed molecular docking trying to explain the obtained results by analyzing plausible structural determinants involved in the interactions with the receptors. We opted for this approach because it can be expected that the separate RYYRIK and cyclic peptide blocks might bind the receptors in different fashions respect to RYYRIK-cyclic peptide conjugates. This seems plausible since individual building blocks are known to bind different receptors: RYYRIK to NOP and the cyclic peptide to KOP.

Comment. Some minor points: Line 124 pKi = 9.29 (no nM)

Answer. this has been corrected

Comment. What would happen if mutations were made on the chimeric peptide? (eg mutating residues on RYYRIK motif or Tyr or cyclic KFFD)

Answer. If we understand correctly, the Reviewer suggests to synthesize more analogs by replacing e.g. Tyr or other amino acids in RYYRIK. We are glad for this good suggestion that will be implemented in future design.

Comment. It would be interesting for the authors to show an overlay of KW-495 and KW-496 binding to KOP. From Figure 5 it seems that the identical cyclic peptide binds to KOP in distinct conformations. This might indicate that the docking results are not reliable. Are the structures in agreement with ref 48?

Answer. We completely agree with this suggestion. Hence, we prepared the new Figure S6 to show the superimposition of KW-495 and KW-496 into the structure of KOP. This overlay predicts that the two conjugates occupy the same cavity of the receptor, albeit in distinct orientations. In the binding pose obtained for the cyclic peptide KW-495, the protonated Tyr1 adopts a classic disposition in the message-binding region of the receptor, while the pose of pose KW-496 is to some extent alternative to the classic one (e.g. JDTic and dynA1−8), but it is not totally unusual for KOP ligands [see ref.55].  In our opinion the possibility that KW-495 and KW-496 might bind to KOP in different conformation, although having the identical cyclic peptide, is not necessarily a symptom of scarce reliability of the computations, taking into consideration that the two compounds exhibited quite different experimental affinities in binding assays.

As for the suggestion to compare the poses to that described in ref 48, we are very grateful to the Reviewer. Indeed, this comment prompted us to observe that only KW-496 shows some similarities with respect to the pose of the cyclic peptide alone from ref 48 (it is important to mention that in the study reported in [ref.48] the cyclopeptide was docked in hMOR). The message portion of the cyclic peptide of KW-496 maintains in KOP a similar disposition to that adopted by the message of the cyclic peptide alone as calculated in MOR [48]. This significant difference might in part explain the higher experimental affinity of KW-496.

In the revised version of the manuscript we utilized the novel PacDOCK web server [reffs.57,58], which allowed to improve the quality of figures 5 and 6.

Comment. Docking of RYYRIK peptide alone should also be performed to validate the docking results of KW-495 and KW-496.

Answer. As already briefly discussed, the docking of the peptide RYYRIK was not carried out, as this compound binds NOP and not KOP. Consequently, it can be assumed that a docking of the RYYRIK alone might lead to different results respect to compounds KW-495 and KW-496.

Comment. What about the docking scores? Are they consistent with the binding data?

Answer. The docking scores of the two peptides were similar; this was not completely unexpected since both conjugates showed affinities in the low nM range. However, since molecular dynamics was not performed, no comments were made on the consistency between docking scores and binding data. The scoring functions represent approximations, which do not consider the entropic free energy term. Besides, the two conjugates have different lengths. Considering the large number of degrees of freedom for the two large compounds, it is difficult to make accurate predictions on binding values from the docking scores.

Reviewer 2 Report

Dear Authors,

                     The article reflects high scientific significance and effort. In my suggestion, it can be improved further, and some parts must be improved. Here are my suggestions

Abstract- Can be improved

Introduction - The experimental part in the introduction must be improved with adequate information. 

Methodology- The MD part is not described well, it must be described properly, and the author should perform a few more steps for analysis like RMSF, H-bonds, free energy calculation and SASA. And they must mention how long the simulation was performed!

Results-

1.      The NMR was making some noise! Can the author explain the other peaks?

2.      MD simulation part is not very clear, it should be tuned with elaboration.

3.      Suggestion for the authors, they must check the peptide-only system for clustering analysis in MD simulation!

4.      NO RMSD or RMSF data!! But they mentioned the simulation performance!

5.      The interaction analysis was done by!

6.      Is it not possible to make the energy profiling for say Gibbs binding free energy?

Discussion- MD Simulation part is not described well; it must be described properly.

Author Response

The article reflects high scientific significance and effort. In my suggestion, it can be improved further, and some parts must be improved. Here are my suggestions

Comment. Abstract- Can be improved

Answer.  As suggested, the Abstract was significantly expended

Comment. Introduction - The experimental part in the introduction must be improved with adequate information. 

Answer: More information on the performed experiments were added to Introduction.

Comment. Methodology- The MD part is not described well, it must be described properly, and the author should perform a few more steps for analysis like RMSF, H-bonds, free energy calculation and SASA. And they must mention how long the simulation was performed!

Answer We apologize for not having explained our protocol accurately. The molecular dynamics of the poses was not carried out, but rather we conducted a re-scoring of the docking poses after a partially restrained equilibration to allow a relaxation of the pose and eliminate any unfavorable clash. We have more adequately implemented the Materials and Methods section. We thank the reviewer for the suggestion, a more in-depth MD study will certainly be the subject of future studies.

This comment applies also to the other Comments related to MD. See also comments to Reviewer 1

Comment. The interaction analysis was done by!

Answer. We utilized Biovia Discovery Studio 2016. This has been clearly indicated in the revised version of the Supporting Information.

Round 2

Reviewer 2 Report

Dear Authors,

Thank you for this nice work and your answer. If MD did not carried out it would be better to remove it.